# Non-Microbiological Mycobacterial Detection Techniques for Quality Control of Biological Products: A Comprehensive Review

**DOI:** 10.3390/microorganisms12040788

**Published:** 2024-04-12

**Authors:** Marine Marius, Clothilde Fernandez

**Affiliations:** Sanofi, 1541 Ave. Marcel Mérieux, 69280 Marcy l’Etoile, France; clothilde.fernandez@sanofi.com

**Keywords:** mycobacteria, mycobacteria detection, contaminants, biological products, biopharmaceutical quality control, nucleic acid amplification techniques, alternative methods

## Abstract

Mycobacteria can be one of the main contaminants of biological products, and their presence can have serious consequences on patients’ health. For this reason, the European Pharmacopoeia mandates the specific testing of biological products for mycobacteria, a critical regulatory requirement aimed at ensuring the safety of these products before they are released to the market. The current pharmacopeial reference, i.e., microbial culture method, cannot ensure an exhaustive detection of mycobacteria due to their growth characteristics. Additionally, the method is time consuming and requires a continuous supply of culture media, posing logistical challenges. Thus, to overcome these issues, pharmaceutical industries need to consider alternative non-microbiological techniques to detect these fastidious, slow-growing contaminating agents. This review provides an overview of alternative methods, which could be applied within a quality control environment for biological products and underlines their advantages and limitations. Nucleic acid amplification techniques or direct measurement of mycobacteria stand out as the most suitable alternatives for mycobacterial testing in biological products.

## 1. Introduction

Mycobacterial contamination of biological products refers to the presence of mycobacterial species such as *Mycobacterium* (*M.*) *tuberculosis*, *M. avium* and others in products derived from biological sources, such as vaccines, cell cultures, blood products and other biologicals. Although most mycobacterial species are nonpathogenic, their survivability and ubiquitous presence across environments make them an ever-present potential source of contamination in eukaryotic cell-based bioproducts [1]. Mycobacteria are characterized by their unique cell structure, which includes a complex cell wall composed of mycolic acids, peptidoglycans, and other lipids [2]. This cell wall structure makes mycobacteria highly resistant, allowing mycobacterial growth in various environments, even including the most hostile ones [3], and making their elimination during the production of biologicals challenging. The opacity of the cell wall further results in a slow bacterial metabolism, with replication times ranging from 2 to 48 h depending on the species. This makes mycobacteria difficult to detect on media cultures.

Contamination of biologicals poses a significant risk to patients’ health considering the potential infections and serious adverse effects [4,5,6]. Such contaminants can be introduced at any stage of production, with sources varying from raw materials and environments to operators. Therefore, the detection and isolation of mycobacteria is a regulatory requirement to ensure the efficacy and safety of biologicals [7].

The reference ‘compendial assay’ method is based on the detection of mycobacteria in biologicals using the microbial culture method. Considering the slow growth of some of the species, e.g., *Mycobacterium tuberculosis* complex (MTBC), culture incubation can be notoriously long and can last for over 56 days. Accordingly, the duration of mycobacterial testing is the longest of all bacteriological quality control (QC) tests for biologicals. It is twice the duration of mycoplasma culture tests and nearly four times the duration of sterility tests. Additionally, the method requires a continuous supply of culture media, which again could be challenging.

Thus, for products with a limited shelf-life, reducing this prolonged duration of process can be critical to ensure their timely and efficient release. These challenges could be addressed using alternative culture-free detection methods, which could drastically reduce the amount of time required and the requirement of raw materials, such as culture media [8,9].

According to the guidelines on the implementation of alternative methods, an alternative candidate method must demonstrate non-inferiority and, ideally, should be superior to the reference method [8,9,10], where superiority means an overall improvement in the product’s safety during early detection, increased sensitivity and/or increased specificity.

The alternative methods that are available are based on different detection mechanisms such as nucleic acid amplification techniques (NATs, e.g., polymerase chain reaction [PCR]), flow cytometry or electrochemical detection. While a considerable number of studies are focused on mycobacterial testing to assist diagnostics in a clinical setting, the information on using mycobacterial testing for the quality control of biologicals is limited. The method deployed in a quality control setting needs not only to be sensitive but also exhaustive enough to enable the detection of the lowest number of cells of any mycobacterial species.

The purpose of this review is to provide direction and information for the improvement of mycobacteria testing in a quality control environment. This review presents different alternative technologies and summarizes their advantages and limitations, with the goal of suggesting a detection method that could potentially replace current compendial mycobacteria testing for the release of biological products.

## 2. Compendial Assay

### 2.1. Compendial Assay: The Gold Standard for Mycobacterial Detection

The microbial culture method or compendial assay is used for detecting mycobacteria in diagnostics as well as for the quality control of biologicals [7,11,12]. Although its limit of detection (LOD) has never been published, it is reported that this method is highly sensitive and can thus detect even the lowest number of mycobacteria. The method can be paired with various identification techniques which can be used for further investigations following a positive result. These identification techniques can be based on either microscopic morphology with Ziehl–Neelsen staining, deoxyribonucleic acid [DNA] sequencing or matrix-assisted laser desorption/ionization time of flight mass spectrometry (MALDI-TOF MS) [12].

Different regulatory texts describe the detection of mycobacteria by culture with varying degrees of details on their use in the quality control of biologicals. European Pharmacopoeia (Eur. Ph.) 2.6.2 requires the product to be assessed on three different media, which includes two solid and one liquid media, in triplicate, for a period of 56 days. If any of these nine samples is positive, i.e., mycobacterial growth is established during the incubation period, the result is considered positive, whereas if none of these media show any mycobacterial growth after the incubation period, the result is considered negative, and the product passes the quality control test [7].

Mycobacteria-specific media exist in both solid and liquid forms and are complementary to each other. Most of these media are developed for mycobacterial detection in sputum, tissues or body fluid samples and contain bacterial and fungal growth inhibitors, which allows the growth of specific target mycobacterial species [12]. Different media are commercially available depending on their applications (Table 1). Overall, liquid media enable an early detection of mycobacterial growth and have a better recovery rate than solid media [12,13]. Egg-based and agar-based media are the two most commonly used solid media. Due to the long incubation period for mycobacterial recovery, solid media are prepared in slant tubes. However, counting and numeration of mycobacterial colonies are extremely difficult due to the limited slant area and non-homogenous spread of colonies [12,13]. Apart from the requirement of two solid and one liquid media, there are no specific recommendations in the Eur. Ph. [7].

### 2.2. Advantages and Limitations of Compendial Assay

The microbial culture method is a simple assay and does not require any expensive equipment or skilled personnel. Besides the low hands-on time, this assay offers a high level of sensitivity, which is of particular importance to detect potential mycobacterial contaminations in biologicals.

However, in a quality control setting, the method’s prolonged incubation time remains a limiting step in the process of releasing biological products. Also, the method is not suitable for the detection of all mycobacterial species, as some of them are not easily cultured on media (e.g., *M. lepraemurinum*), or have specific nutritional requirements (e.g., *M. haemophilum*), whereas some do not grow on media at all, (e.g., *M. leprae*) [13]. Finally, and despite their selectivity, these media are not specific to mycobacteria and can support the growth of other actinomycetes such as *Nocardia*. Although unlikely in a quality control context, this can, in theory at least, lead to false positive results [11].

## 3. NAT: Alternative Mycobacterial Detection Methods

### 3.1. Overview

Over the past 20 years, molecular biology methods for the detection, isolation and identification of mycobacteria have developed significantly. These methods not only avoid lengthy culturing periods and incubations but can also render results in few hours. In addition to the diagnosis of mycobacterial infections (MTBC or nontuberculous mycobacteria), numerous applications have been developed using nucleic acid detection as the marker of bacterial, viral or fungal contamination or infection [14]. These include the detection of non-cultivable microorganisms [15], rapid diagnostics [16], environmental contaminant search [17,18,19] and food safety [20]. NATs can also be used to test for the presence of adventitious agents in biological products [21]. As outlined in the Eur. Ph. 5.1.6, NATs can be used as an alternative method for the detection of microorganisms [9]. Expending on the Eur. Ph. 2.6.21, NATs could represent a suitable alternative to the culture method for detecting mycobacterial contaminants in the quality control of biological products, provided that such an assay is validated [22,23,24,25,26].

Before introducing the different NAT-based approaches for mycobacterial detection in Section 3.3 and Section 3.4, it is important to present an overview of the main nucleic acid extraction methods available. Indeed, the lysis of the mycobacterial cell wall and the extraction of nucleic acids from the sample are required prior to amplification.

### 3.2. Extraction Methods

The sensitivity and specificity of NATs depend on both the efficiency of the nucleic acid extraction method and the quality of extracts. The purity level of extracts is equally important to avoid the presence of PCR inhibitors.

Different extraction methods are documented. However, the choice of the extraction method, and of the extraction protocol, will depend on several factors, such as the nature, size and number of samples to be tested. The optimization of these techniques is crucial for the success of microbial detection and identification [2,27,28,29,30,31].

#### 3.2.1. Nucleic Acid Extraction Methods

Compared with other commercially available kits or automatized extraction methods, organic solvent extraction is the most sensitive and efficient nucleic acids extraction method available [27,30,31,32,33], and the phenol–chloroform method is considered as the gold standard for mycobacterial nucleic acids extraction [34]. As the method involves the use of volatile and hazardous chemicals, there is a need for specialized equipment and skilled personnel. Moreover, the method is not suitable for the quality control testing of all biologicals due to its limitation in assay standardization and throughput [35,36].

Spin column extraction is a solid phase extraction method. It is widely used in molecular biology and is effective in recovering even minute amounts of DNA with a high level of purity [36,37].

Magnetic beads extraction is based on the selective binding of DNA to magnetic nanoparticles. The DNA–magnetic bead complexes can then be isolated from the lysate using an external magnet [36]. This method can be used for mycobacterial DNA extraction [38]. Both spin column and magnetic beads extraction methods are easier to standardize and/or automate than the phenol–chloroform method, and also require less skilled workers.

Mycobacterial nucleic acids can also be extracted without any downstream purification steps. This approach is usually adopted while handling fresh cultures. However, it cannot be used for mycobacterial detection in biologicals due to the low number of cells expected and the complexity of the matrix.

Based on the current literature, extraction with magnetic beads and extraction with a spin column appear to be the two most appropriate techniques for quality control testing biologicals. They do not involve using hazardous chemicals and can also be automated to provide quick and reproducible results.

#### 3.2.2. Membrane Lysis Methods

For the selection of a cell lysis technique, the samples to be assessed and the nucleic acids extraction method to be used are key parameters [31]. Different methods can be combined to increase the efficiency of cell lysis. Mechanical lysis is the simplest and one of the most efficient techniques to break down mycobacterial cell walls [18,39]. The most commonly used mechanical lysis methods include bead beating [31,40], boiling [41,42] and sonication [40].

A chemical lysis of a mycobacterial cell wall can be performed with cetyltrimethylammonium bromide, which induces cell lysis by precipitation of cell wall components such as free lipoglycans and polysaccharides. However, the technique needs to be coupled with organic solvent extraction (phenol–chloroform), and thus requires specialized equipment and skilled personnel [29,39]. Detergents such as sodium dodecyl sulphate and Triton X-100 can be used to dissolve the bacterial cell wall [30]. Several kits with lysis buffers containing these detergents are commercially available.

Another lysis method is enzymatic lysis, which is often combined with other membrane lysis methods. Proteinase K induces lysis by hydrolyzing peptide bonds of proteins. This enzymatic digestion allows the purification of DNA extracts. The method is frequently used in nucleic acids extraction protocols [19,27,30,36] and is often combined with lysis buffers as it can withstand a range of pHs, temperatures, and detergents. Lysozymes have a specific lytic activity on the peptidoglycan (PDG) layer of the bacterial cell wall. Once the PDG layer is exposed by a prior lysis method, it becomes accessible to lysozymes for further lysis [30,31].

The selection of a method for mycobacterial cell wall lysis depends on the sample type (e.g., master seed lot and cell cultures for vaccine production) and the purpose of the test [27,31]. In clinical diagnostics, the mycobacterial load in patients’ samples (sputum, blood, body fluids, exudates from abscess, tissues from biopsies) is generally higher and membrane lysis of only a fraction of mycobacterial cells will still yield enough genetic material for detection. Whereas for biologicals, the test should be able to detect even the lowest number of mycobacterial cells in product samples (cell cultures, bulk harvest). Thus, for the quality control of biologicals, different lysis methods can be combined to increase the efficiency of extraction [19].

### 3.3. PCR Techniques

There are several techniques based on the principle of PCR for the quality control of biologicals. Most of these techniques were developed for the diagnosis of *M. tuberculosis* and helped to reduce the time to result, thus allowing for a faster initiation of appropriate treatments [14,43,44,45].

#### 3.3.1. Gene Target

The choice of a gene target is particularly important as it defines the specificity of amplification and, accordingly, the reliability and exhaustivity of the results. Many PCR methods rely on the amplification of the 16S ribosomal RNA (16S rRNA) gene as it contains both regions conserved across the mycobacterium genus and species-specific variable domains [43,46]. Thus, depending on the 16S rRNA gene region targeted by the primer, this approach can be used either to detect the presence of mycobacteria or to identify specific mycobacterial strains or species [43,47,48,49]. Additionally, the *rpoB* gene [50] and the 16S-23S rRNA gene spacer region are also used as targets for the amplification of mycobacterial DNA [51]. The gold-standard target gene used for the detection of tuberculosis in clinical settings is the 16S rRNA gene [48,49]. A different mycobacterial gene, *hsp65*, has also been used to detect a broad range of mycobacteria species in water samples (based on the *hsp65* gene) [18].

#### 3.3.2. Amplification

Once the target gene is selected and the corresponding primer is available, the nucleic acids extract can be amplified using PCR [52]. Several PCR methods, such as real-time PCR, nested-PCR or touchdown PCR, are widely utilized in medical applications. These techniques are highly valued for their ability to reduce the time to result, enabling the early initiation of appropriate treatment in diagnostic settings [14,43,44,45]. Touchdown PCR is a variation of the classic PCR method with an improved specificity. It is based on the principle that a higher hybridization temperature entails a more specific pairing between the primer and template. Thus, in the first steps of touchdown PCR, only the regions with the highest specificity are amplified, thus limiting the amplification of non-specific products. In the following cycles, as the hybridization temperature is incrementally reduced, the target products will be further amplified and can outcompete any remaining non-specific products [53]. The specificity of touchdown PCR makes it particularly relevant for differentiating between strains in mycobacterial diagnosis [54].

Nested-PCR is designed to increase the specificity and sensitivity of detection. It does so by amplifying a first target template, one that could be either rare or, on the contrary, that yields a range of amplification products if the selected pair of primers has different binding sites. This approach can be especially useful in the context of high background extraction samples. Then, using a second pair of ‘nested’ primers (i.e., primers targeting a region within the initial template), the second run amplifies the target region, resulting in a higher specificity. Using the 16S rRNA gene, nested-PCR approaches have been developed to assist in the diagnosis of both tuberculous and nontuberculous mycobacterial infections [55].

#### 3.3.3. Endpoint PCR Detection

There are several approaches for the detection of mycobacterial nucleic acids following amplification. The first and oldest method involves the migration of the amplicons onto an agarose gel followed by their detection by non-specific a double strand intercalating agent like ethydium bromide [46]. This technique is reliable for checking the purity of DNA extracts and is often coupled with other detection methods for diagnostics. Additionally, to further improve the detection of specific amplicons, hybridization probes can be used. These probes consist of short oligonucleotides (usually 13–25 nucleotides long) labelled with a fluorescent dye. After PCR amplification, the amplified DNA can be hybridized on a microarray spotted with an array of specific oligonucleotide probes [43]. However, considering the handling time of post-amplification analyses and the risk of contamination, endpoint PCR techniques are not suitable to be used in a quality control setting (Table 2). Also, these techniques are focused on diagnostics, and their lack of exhaustivity and sensitivity is prohibitive for mycobacterial testing of biologicals.

#### 3.3.4. Real-Time PCR

Real-time PCR has been a breakthrough in molecular biology technology. The technique is based on the detection of amplified products generated during PCR cycles. As this technique avoids post-PCR manipulations, it limits the risk of the amplified products being contaminated, while also reducing the time to result. Most real-time PCR instrumentation can analyze 96- or 384-well plates, which further improves the analysis throughput, combined with an automatization of the technique [16]. Other advantages include an increased sensitivity and specificity for the target DNA (Table 2). However, the detection of amplified products using DNA intercalator dye such as SYBR^®^Green is not specific [56]. The detection of mycobacteria can be achieved with the use of specific probes [18].

Due to its ability to quantify the amount of DNA in analyzed samples against a range of calibrated reference dilutions, real-time PCR is also known as quantitative PCR (qPCR). This technology has broadened possibilities in the field of mycobacterial detection. In addition to diagnostics, mycobacterial detection by real-time PCR can also be used for environmental analysis, e.g., water quality testing [17,18,57]. Real-time PCR is also used for testing other bacterial or viral contaminants [21]. Compared to other PCR-based approaches, this technique offers several advantages such as increased sensitivity, specificity, efficiency and exhaustivity (i.e., the capacity to detect broadest possible number of mycobacterial species) while testing biologicals (Table 2). However, efficiency and sensitivity remain dependent on the nucleic acids extraction method selected, whereas the specificity and exhaustivity depend on the design of primer and probe.

#### 3.3.5. Digital PCR

Digital PCR (dPCR) is an endpoint PCR technique that allows an absolute quantification of a sample’s DNA content. It is based on the partitioning of a sample into numerous independent PCR reactions such that each partition contains either a few or no target sequences [58]. This technique requires a post-amplification data analysis, which is more time-consuming than real-time PCR (Table 2).

dPCR and real-time PCR are often compared to each other. Owing to its design (specifically the partitioning mechanism), the sensitivity of dPCR for the detection of contamination is lower than that of real-time PCR, although it has a better limit of quantification [59]. While quantification is not required for mycobacteria testing, a particularly high sensitivity is crucial. Given that the application of dPCR in this context has not yet been reported in the literature, its potential application in this area warrants assessment.

### 3.4. Next-Generation Sequencing

The advent of next-generation sequencing (NGS) techniques brings a new opportunity for the detection of mycobacterial nucleic acids [60,61]. The limitations of previous generation sequencing techniques [62] did not allow for the use of sequencing as a routine method for mycobacterial detection. First-generation sequencing is usually reserved for genetic studies. Now, NGS techniques can be an interesting alternative as these are faster, easier to perform and more financially accessible than previous-generation technologies [63].

Like other mycobacterial nucleic acids detection techniques, NGS techniques have diagnostic applications. The technology is useful for identifying the species or strains involved in a disease and can provide information about possible drug resistances [61].

The NGS technology can detect all species of mycobacteria within a sample. However, such indiscriminate sequencing can generate a vast amount of sequencing data, including a potentially large number of irrelevant sequences which would further complicate data management and analysis; therefore, the time to result is longer (Table 2). Furthermore, these analyses often require high-capacity instrumentation which comes at a higher cost. The development of non-whole sequencing techniques targeting specific mycobacterial DNA region could lead to cost effective solutions. However, NGS-based techniques still need to be assessed for their sensitivity, specificity and exhaustivity in the context of testing biologicals for mycobacteria during the quality control process.

### 3.5. Choice of NATs

NATs have rapidly evolved from the detection of amplicons on agarose gel to NGS in the span of a few years, as have their applications. These methods have the fact that they allow us to bypass the lengthy process of mycobacterial culture in common, which can significantly reduce the time to result. However, it is worth noting that nucleic acid detection does not provide any information about the viability of contaminants.

A comparison of NATs is presented in Table 2. Among all the NATs, real-time PCR with mycobacteria-specific probe detection appears to be the most suitable technique for mycobacterial testing. The method is not only rapid (low handling time and reduced time to result) but also offers a high sensitivity and specificity. As the method does not require any post PCR analyses, it also limits the risks of the amplified products becoming contaminated. However, to our knowledge, no applications of real-time PCR have been described in the literature for the quality control of biologicals.

## 4. Other Methods for Mycobacterial Testing

### 4.1. Protein Detection by HPLC or MALDI-TOF MS

Protein detection methods are widely used for the identification of mycobacterial species for diagnostic purposes. The development of “protein identity cards” allowed individual mycobacteria species to be identified. One of the first techniques developed based on this approach was high-performance liquid chromatography (HPLC) [64]. HPLC targets mycolic acids from the cell wall for the identification of mycobacterial species [64,65]. More recently, MALDI-TOF MS protocols have also been used for mycobacterial identification [66]. The MS analysis is based on the principle that each individual mycobacterial species has a unique spectrum [67]. It is important to note that the quality and performance of a detection test based on this method is highly dependent on the protein extraction protocols [11]. Moreover, the main aim of techniques based on the detection of mycobacterial proteins is the identification of mycobacterial species for diagnostic purposes. In the context of the quality control of biologicals, detection is the main purpose, and the identification of species is rarely required, except in cases of investigation with a proven contaminant. As there is not yet a spectrum for each mycobacterial species, an exhaustive detection cannot be performed using HPLC or MALDI-TOF MS analyses.

### 4.2. Viable Mycobacteria Detection

Cytometry techniques can be used for the detection and numeration of both viable and non-viable bacteria cells, depending on the selected labels [68]. All cytometry techniques involve labelling cells with fluorescent dyes, scanning each cell with a laser, and collecting the emission levels of each fluorescent label for each cell. This can be carried out with either a circulating liquid phase (flow cytometry) or a solid phase (scanning cytometry).

Flow cytometry allows for the detection and numeration of mycobacteria in real-time. Due to the variations in labeling strategies, the difference between live and dead mycobacteria can also be calculated without preparing any cultures [69]. While some labels bind to the cell DNA and highlight the damaged membranes [70], others can specifically recognize intact membranes using a labelled antibody or labels for esterase activity [69,70]. Flow cytometry numeration is precise and accurate [71]. However, it is generally used for the preparation of reference strains and diagnostic applications, especially when the concentration of cells is expected to be high.

In solid-phase cytometry, samples presumed to contain mycobacteria are filtrated onto a solid membrane. The filtration is followed by labelling of the bacteria. Fluorescence signals can be collected using laser-based scanning cytometry of the solid phase surface [72] or a charge-coupled device (i.e., a CCD camera) [73]. Mycobacteria can be gated depending on their size and fluorescence and differentiated from background noise.

Overall, cytometry techniques are interesting candidates for quality control of biologicals as they allow for a direct measurement of samples. However, they lack some of the key features required to qualify as an alternative method for mycobacterial testing. The main drawback of cytometry techniques is their lack of sensitivity and, for this reason, they are mostly used on samples containing a high number of mycobacteria, such as cultures [71] or biological samples for diagnostics [69]. Moreover, if the mycobacterial testing is performed on samples with a complex matrix, the matrix composition can generate background fluorescence which will hinder the interpretation of results.

### 4.3. Electrochemical Detection

The electrochemical detection technique is based on the measurement of the acetyltransferase activity of the Antigen 85 (Ag85) complex [74]. This complex is involved in the construction of the mycobacterial cell wall. The enzymatic activity of this species-specific protein complex is determined by voltametric measurement. As both substrate and products can be detected as separate peaks, their presence confirms the Ag85 enzymatic activity, and in turn, the presence of viable mycobacteria.

The main advantage of this technique is that it allows for the direct detection of viable mycobacteria. However, samples cannot undergo a real-time analysis as it requires a 24 h incubation period. It is important to note that the duration of incubation has been set to include a range of mycobacterial species and it might not be adequate to detect the enzymatic activity of all species. In any case, the technique is not expected to be exhaustive as some species require their own specific culture conditions (i.e., not adapted for intracellular mycobacteria species, such as *M. leprae*). Finally, the method has an LOD around 10^3^–10^4^ cells/mL and thus lacks the required level of sensitivity for quality control of biologicals.

### 4.4. Immunodetection

Antigens specific to a mycobacterium can be targeted by an adapted antibody. Nour-Neamatollahi et al. (2018) developed a rapid and simple tool for the point-of-care diagnosis of tuberculosis, allowing mycobacteria to be detected directly in a patient’s sputum [75]. This method involves the filtration of treated samples, followed by the visualization of trapped bacilli on the filter using an immuno-chromatographic method with a gold conjugate. While it has been shown to be as sensitive as the Ziehl–Neelsen method in a clinical context, it is not applicable in a quality control setting due to its limitations. The main drawbacks include its low sensitivity (3 × 10^3^ cells/mL) and exhaustivity.

## 5. Implementation of an Alternative Method for Biopharmaceutical QC Control

### 5.1. Validation of Alternative Method for Release Testing

Before being implemented in a quality control setting, a potential alternative method must undergo validation, as described in different regulatory guidelines such as the Eur. Ph. (Chapter 5.1.6) [9] the United States Pharmacopoeia ([USP] Chapter <1223>) [10] and the PDA technical report 33 [76].

According to these regulatory requirements, an alternative method must have its efficiency demonstrated. In short, it must be at least comparable to the compendial assay and, ideally, show superiority over it. Validation criteria depend on the type of microbiological tests. In any case, the alternative method must bring a benefit compared to the compendial assay to justify the replacement.

### 5.2. Requirements for Techniques Compliant with Biopharmaceutical QC Control

The current compendial assay for mycobacterial testing of biological products is simple and sensitive. However, it requires a 56-day incubation time and can be one of the time-limiting steps in the release process of biological products. It is also not exhaustive as it cannot detect non-cultivable mycobacteria.

Regulators have been encouraging the pharmaceutical industry to develop and implement new detection technologies. As stated in the General Notices and Requirements in the USP, “Alternative methods and/or procedures may be used if they provide advantages in terms of accuracy, sensitivity, precision, selectivity, or adaptability to automation or computerized data reduction, or in other special circumstances” [10]. Therefore, the ideal alternative method for mycobacterial testing in a quality control environment should be simple, rapid, sensitive and specific. Ideally, this method should also have a high throughput and allow as much automation as possible. These requirements are discussed in turn in Section 5.2.1, Section 5.2.2, Section 5.2.3, Section 5.2.4, Section 5.2.5 and Section 5.2.6.

Most new alternatives techniques available offer a faster time to result and improved exhaustivity. These advantages highlight the potential of these alternative methods as candidates for replacing the current compendial assay. The main characteristics of the methods discussed in this review are outlined at the end of this section (Table 3).

#### 5.2.1. Simplicity

The criterion of simplicity is defined by the requirements of instrumentations/operator expertise, and the ease of use. A test should not require any specific qualifications for operators, or expensive materials, and should also be easy to perform. The compendial assay is a simple test. Similarly, electro- [74] or immunodetection methods [75] rely on simple ready-to-use equipment, whereas PCR, NGS, MALDI-TOF MS approaches and cytometry methods require specific expensive equipment and skilled operators.

#### 5.2.2. Time to Result

The time to result is another challenge faced with the compendial assay due to the long incubation period. Most of the proposed alternative methods have much shorter time to results, with results available within a few hours to 2 days, except for the NGS method, which may need longer due to data analysis.

#### 5.2.3. Sensitivity

With regard to mycobacterial detection, the sensitivity of any technique corresponds to the lowest detectable concentration of viable bacteria. It is defined by the compendial assay, and an alternative technique needs to be at least as sensitive as the reference technique. However, it is important to note that the compendial assay’s sensitivity involves a degree of variability as the growth time of mycobacteria differs from one species to the other, ranging from 2 days to 8–12 weeks. The real-time PCR is the most sensitive NAT approach, but it also depends on the efficiency of extraction. The NGS’ sensitivity must still be reported and there are difficulties in detecting low contamination levels in complex samples. The mycobacterial protein detection by HPLC or MALDI-TOF MS depends on the quality of protein purification. In the current literature, there are no data or reports on the sensitivity of cytometry, electro-detection, or immunodetection for mycobacteria detection. Based on the principle of these techniques, it is likely that they are not sensitive enough.

#### 5.2.4. Specificity

Specificity is the ability of the test to detect mycobacteria among a range of different microorganisms [10]. Ideally, a positive result should be coming from a mycobacterial contamination without any nonspecific interferences. In theory, NATs are highly specific due to their primer design and their specificity can be increased with probe detection. Other techniques can also be specific depending on the test design and the samples to be assessed.

#### 5.2.5. Automation

Automation can improve the efficiency of tests (cost reduction, early results, reproducibility). It also offers an advantage by reducing the operator’s handling time, thereby reducing the required full-time equivalents. Extraction of nucleic acids and amplification can be automated; however, endpoint analysis cannot be automatized, thus making it semi-automated. Automatized instruments are available for the HPLC, MALDI-TOF MS and cytometry approaches. However, direct measurement techniques such as electro-detection and immunodetection cannot be automated.

#### 5.2.6. Throughput

A method’s throughput determines the number of samples that can be analyzed at the same time. For real-time PCR and dPCR with an automatized extraction, the number of samples that can be analyzed at the same time is defined by the plate capacity (from 96, up to 384 samples). Additionally, depending on the technology selected, multiple plates can be analyzed at the same time to increase throughput. As expected, analyzing amplicons on agarose gels has a much lower output.

Data management and analysis can also be a limiting factor for the processing of a large number of samples. For instance, protein analysis by HPLC or NGS analysis generates a significant amount of data which, in turn, can be difficult to analyze. Although the throughput of analysis can be improved with new and more performant software, it is by no mean sufficient to reach the pace of real-time PCR and dPCR [77]. Cytometry techniques allow analysis in 24- or 96-well plates, but their use case must be assessed for the detection of mycobacteria [78]. A direct analysis with electro-detection or immuno-detection cannot be automatized, but their short turnaround time allows the procedure to be repeated perform multiple analyses. However, even these methods cannot reach the throughput levels of real-time PCR and dPCR.

**Table 3 microorganisms-12-00788-t003:** Comparison of alternative methods.

	Criteria	Simplicity	Time-to-Result <2 Days	Sensitivity ≤10 Cells/mL	Specificity	Automation	Throughput	Reference
Methods								
Compendial assay		√	✕	√	√	✕	✕	[7,11,12]
Ideal alternative method		√	√	√	√	√	√	
**Available alternative methods**								
PCR endpoint		√	√	✕	√	√✕ *	✕	[42,43,44,46,48,51,52]
Real-time PCR		✕	√	√	√	√	√	[16,17,18,21,41,44,45,49,52,56,79,80,81]
dPCR		✕	√	✕	√	√	√	[21,58,59]
NGS		✕	✕	NBA **	√	√	✕	[14,16,21,60,61,63]
HPLC		✕	√	NBA **	NBA **	√	✕	[64,65,77]
MALDI-TOF/MS		✕	√	NBA **	NBA **	√	✕	[66,67]
Cytometry		√	√	✕	NBA **	√	NBA **	[68,69,70,71,72,73,78]
Electro detection		√	√	✕	NBA **	✕	✕	[74]
Immunodetection		√	√	✕	√	✕	✕	[75]

dPCR, digital polymerase chain reaction; HPLC, high-performance liquid chromatography; MALDI-TOF/MS, matrix-assisted laser desorption/ionization time-of-flight mass spectrometry. * Semi-automated technique ** NBA: Needs to be assessed.

## 6. Conclusions

Based on this review, NATs combined with an efficient extraction method appears to be the most suitable alternative to the compendial assay for mycobacterial testing in biological products. Real-time PCR seems to be the closest to the ideal alternative method. To our knowledge, no other NATs method is available for such an exhaustive, specific and sensitive detection of mycobacteria in biologicals. Additionally, real-time PCR provides results in a few hours, leading to significant reduction in the time to result as compared to the compendial assay. The development of this type of technique could be the best alternative method.

In any case, the prior validation of an alternative method will lead to the potential replacement of the compendial assay. It is important to note that even if a NAT method has the potential to overcome some of the limitations of the compendial assay, it most likely will come with its own set of drawbacks. Therefore, the choice of an alternative method should be carefully made and, ideally, integrated into an overall contamination control strategy.

## Figures and Tables

**Table 1 microorganisms-12-00788-t001:** Different commercial media.

Liquid Media	
Proskauer and Beck medium	Not available commercially; Needs to be prepared in-house based on empirical formulation; Preparation process is difficult and can induce variability between batches.
Sauton medium	
Kircher medium	Used for the diagnosis of tuberculosis.
Dubos medium	Allows the development of most cultivable mycobacteria.
Middlebrook medium	Middlebrook 7H9 is one of the commonly used media for the recovery of all mycobacterial species.
**Solid Media**	
Lowenstein–Jensen medium	Most commonly used and oldest egg-based medium; Selective; inhibits bacterial and fungal growth. Strongly supports the growth of MTBC.
Middlebrook 7H10 and 7H11 media	Standard agar-based media; Mycobacterial growth time on these media is shorter than that on egg-based media but longer than that on liquid media; Expensive and have a short storage life.

MTBC, Mycobacterium tuberculosis complex.

**Table 2 microorganisms-12-00788-t002:** Comparison of NAT.

Techniques	PCR Endpoint		Real-Time PCR		Digital PCR		NGS
Detection	ICA	Probes	ICA	Probes	ICA	Probes	
Sensitivity	NO		YES		YES		*
Specificity	NO	YES	NO	YES	YES	YES	YES
Handling time	1–2 h	3 h	30 min	30 min	1 h		4 h
Time to result	3 h	4 h	4 h	3 h	4–5 h		>2 days
Amplified production contamination risk	YES		NO		NO		YES
Automation/high throughput analysis capacity	NO		YES		YES		YES

ICA: intercalating agent of DNA like ethidium bromide, SYBR^®^ Green etc.; NAT, nucleic acid amplification technique; NGS, next-generation sequencing; PCR, polymerase chain reaction. * Must be assessed.

## Data Availability

Not applicable.

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
