# Peer review of "Non-Microbiological Mycobacterial Detection Techniques for Quality Control of Biological Products: A Comprehensive Review"

_microorganisms, 2024, doi:10.3390/microorganisms12040788_

Round 1

Reviewer 1 Report

Comments and Suggestions for Authors

The manuscript titled "Testing of Biological Products: A Review of Alternative Non-Microbiological Techniques for Mycobacterial Testing" summarizes data regarding the detection of several Mycobacteria species. While there are numerous reviews focused on M. tuberculosis, this work is distinguished by its broader focus on various Mycobacteria, making it a valuable contribution to the field. However, some issues need addressing.

Comments:

  1. 1. I suggest the authors make the title clearer.
  2.  
  3. 2. Abstract: 2.1 The sentence "Thus, specific testing is necessitated by regulations..." could be rephrased for clarity. 2.2 The sentence "[...], which could be further challenging..." also requires rephrasing for better understanding.
  4.  
  5. 3. The Introduction would benefit from an additional paragraph detailing the Mycobacteria cell structure and its relevance to the challenges in disinfection and detection.
  6.  
  7. 4. In Section 3.1.2 (Lines 223-230), please clarify the most effective lysis methods for the cases mentioned.
  8.  
  9. 5. Section 3.1.3 contains redundant information with Sections 3.1.1 and 3.1.2; please address this overlap.
  10.  
  11. 6. In Section 3.2.2, it is important to include references and detailed explanations about the applications of various PCR techniques for Mycobacteria detection, rather than merely describing the techniques.
  12.  
  13. 7. Section 3.4 requires references for the statement "The detection of Mycobacteria can be achieved by the use of specific probes."
  14.  
  15. 8. Please provide more details on the application of the methods discussed in Section 3.2.5 for Mycobacteria detection.

Author Response

RESPONSES TO REVIEWER 1 COMMENTS

 General comment:

The manuscript titled "Testing of Biological Products: A Review of Alternative Non-Microbiological Techniques for Mycobacterial Testing" summarizes data regarding the detection of several Mycobacteria species. While there are numerous reviews focused on M. tuberculosis, this work is distinguished by its broader focus on various Mycobacteria, making it a valuable contribution to the field. However, some issues need addressing.

Response: Thank you for the thorough review and valuable inputs, which will enhance the quality and impact of our manuscript.

 Specific comments

  1. I suggest the authors make the title clearer.

Response: Thank you for the suggestions. We have modified the title to “Non-microbiological mycobacterial detection techniques for quality control of biological products:  A comprehensive review”

  1. Abstract: 2.1 The sentence "Thus, specific testing is necessitated by regulations..." could be rephrased for clarity.

Response: The original statement:

“Thus, specific testing is necessitated by regulations such as the European Pharmacopoeia.”

Was expanded and rephrased as follows:

“For this reason, the European Pharmacopoeia mandates the specific testing of biological products for mycobacteria, a critical regulatory requirement aimed at ensuring the safety of these products before they are released to the market.”

2.2 The sentence "[...], which could be further challenging..." also requires rephrasing for                better understanding.

Response: The original statement:

“Additionally, the method is time-consuming and requires a continuous supply of culture media, which could be further challenging.”

Was rephrased as follows:

“Additionally, the method is time-consuming and requires a continuous supply of culture media, posing logistic challenges.”

  1. The Introduction would benefit from an additional paragraph detailing the Mycobacteria cell structure and its relevance to the challenges in disinfection and detection.

Response: This is a very good suggestion as it provides the reader with some background information highlighting the challenges of mycobacteria detection. We have added the paragraph detailing the Mycobacteria cell structure, as suggested, in the introduction section:

Mycobacteria are characterized by their unique cell structure, which includes a complex cell wall composed of mycolic acids, peptidoglycans, and other lipids [2]. This cell wall structure makes mycobacteria highly resistant, allowing mycobacterial growth in various environments, including the most hostile [3], and making their elimination during the production of biologicals challenging. The opacity of the cell wall further results in a slow bacterial metabolism, with replication times ranging from 2 to 48 hours depending on the species. This makes mycobacteria difficult to detect on media culture.

  1. In Section 3.1.2 (Lines 223-230), please clarify the most effective lysis methods for the cases mentioned.

Response: The point we wanted to make has been clarified. Here, we did not want so much to highlight the most efficient lysis method; this will depend on the type of sample. Rather, we wanted to underline that while lysis efficiency is not so much an issue in clinical settings due to the high number of bacterial cells, it is critical in the context of quality control. The paragraph has been updated as follows:

“The selection of method for mycobacterial cell wall lysis depends on the sample type (e.g., master seed lot and cell cultures for vaccine production) and purpose of the test [28,32]. In clinical diagnostics, the mycobacterial load in patients’ samples (sputum, blood, body fluids, exudates from abscess, tissues from biopsies) is generally higher and membrane lysis of only a fraction of mycobacterial cells will still yield enough genetic material for detection. Whereas for biologicals, the test should be able to detect even the lowest number of mycobacterial cells in product samples (cell cultures, bulk harvest). Thus, for the quality control of biologicals, different lysis methods can be combined to increase the efficiency of extraction.

  1. Section 3.1.3 contains redundant information with Sections 3.1.1 and 3.1.2; please address this overlap.

Response: Thank you for noticing it. In order to avoid the overlap, we have removed the entire section 3.1.3

  1. In Section 3.2.2, it is important to include references and detailed explanations about the applications of various PCR techniques for Mycobacteria detection, rather than merely describing the techniques.

Response: This section is now numbered Section 3.3. As per the suggestions, we have added additional information on the applications of various PCR techniques for Mycobacteria detection. These applications are mostly relevant for clinical settings. We have added mentions to 2 additional applications of gene targets in Section 3.3.1:

“The gold standard target gene used for the detection of tuberculosis in clinical settings is the 16S rRNA gene [49,50]. It has also been shown to detect a broad range of mycobacteria species in water samples (based on hsp65 gene) [18].”

In Section 3.3.2, we have also added a sentence highlighting the usefulness of touchdown PCR in the identification of mycobacterial strains:

“The specificity of touchdown PCR makes it particularly relevant for differentiating between strains in mycobacterial diagnosis [55].”

  1. Section 3.2.4 requires references for the statement "The detection of Mycobacteria can be achieved by the use of specific probes."

Response: Thank you for the suggestion. We have cited the following reference to the above-mentioned statement:

Radomski, N.; Lucas, F.S.; Moilleron, R.; Cambau, E.; Haenn, S.; Moulin, L. Development of a real-time qPCR method for detection and enumeration of Mycobacterium spp. in surface water. Applied Environmental Microbiology 2010, 76, 7348-7351.

  1. Please provide more details on the application of the methods discussed in Section 3.2.5 for Mycobacteria detection.

Response: The section referred to is now numbered 3.3.5. Due to overall complexity of the digital PCR principle, we did not provide much additional technical details. However, we added a paragraph drawing a short comparison with real-time PCR. We believe this will allow the reader to have a general understanding of the advantages/limitations of digital PCR compared with real-time PCR. The following text was added:

“dPCR and real-time PCR are often compared to each other. Owing to its design (specifically, the partitioning mechanism), the sensitivity of dPCR for the detection of contamination is lower than that of real-time PCR, although it has a better limit of quantification [60]. While quantification is not required for mycobacteria testing, highest sensitivity is crucial. Given that the application of dPCR in this context has not yet been reported in literature, its potential application in this area warrants assessment.”

Reviewer 2 Report

Comments and Suggestions for Authors

The overall composition of the manuscript is good. The paper is scientifically and methodologically accurate. This manuscript will interest many readers.

However, my recommendation is 'Minor Revision'. More detailed comments are given below.

General comments

The review appears to be a technical description of the methods used in amplifying nucleic acids; however, it needs more scientific evidence of where NAT techniques have been applied for detecting mycobacteria in quality control of biologicals, for example. Therefore, it is suggested that the authors expand the information that refers to the detection of mycobacteria in the quality control of biologicals. It is also suggested that they discuss the advantages of each of the techniques applied to the detection of mycobacteria in quality control in biologicals.

By adding the requested information, the authors' conclusion would be scientifically supported since they conclude that: “Real-time PCR seems to be the closest to the ideal alternative method. To our knowledge, no other NATs method is available for such an exhaustive, specific, and sensitive detection of mycobacteria in biologicals. Additionally, real-time PCR provides results in few hours leading to significant reduction of time to result as compared to the compendial assay”.

Specific comments

1. Authors need to reinforce their focus on why they carry out this work. The way it is presented could be stronger. The authors must indicate which strategies were used to achieve their objectives.

2. It is suggested that the authors provide more information about their references throughout the main document. The way they are presented is only descriptive, and there is no comparison between them as one would expect in a review.

3. A short introduction and discussion in each section must be provided before describing all the results that support each section. This is to help the reader understand the importance of the results.

4. For a better understanding of readers. It is suggested that authors include figures in their main document.

5. Please change fastidious by large or difficult. Revise the main document.

6. 2.2. . Advantages and limitations of compendial assay. Revise and remove the second point.

7. 3.1.3. Choice of nucleic acids extraction method for quality control of biologicals. This paragraph is repetitive. Please remove it.

8. 3.1. NATs. This section contains Nucleic acid extraction methods and Membrane lysis methods. These two subsections are not corresponding with NATs. Please rename the title of this section.

9. 3.2.1. Gene target: Please add examples that have been detected with this technique.

10. It is suggested that the authors describe Table 2 in the main document.

11. 4.4. Immunodetection section More information should be provided in this section.

12. 5.2. Comparison of techniques compliant to biopharmaceutical QC control. Please provide a better explanation. The way it is written is abstract.

13. Table 3. Alternative methods comparison. For a better understanding of readers, it is suggested that the authors list the methods mentioned in the main document. Why some are marked in blue and bold? Additionally, add the appropriate references in a column.

Author Response

Sylvia Shi

Assistant Editor

Special issue of Microorganisms (Detection and Identification of Pathogenic Bacteria and Viruses)

Dear Sylvia Shi,

I am writing to submit the revised version of our manuscript titled “Testing of biological products: A review of alternative non-microbiological techniques for mycobacterial testing” for consideration in Microorganism. We appreciate the feedback provided by the reviewer and have carefully addressed each comment in our revised draft. The suggestions received were thoughtful and helped us to make the manuscript more impactful. Below, please find our detailed responses to the reviewer comments.

We believe that the revisions have significantly improved the quality and clarity of our work and will contribute valuable insights to the field.

We kindly request that you consider our revised manuscript for publication in Microorganism.

Thank you for your time and consideration. We look forward to hearing from you soon.

Round 2

Reviewer 1 Report

Comments and Suggestions for Authors

My comments have been addressed by the authors.